# COVID-19 stress syndrome in the German general population: Validation of a German version of the COVID Stress Scales

Stefanie M. Jungmann[1]*, Martina Piefke[2], Vincent Nin[2], Gordon J. G. Asmundson[3], Michael Witthöft[1]

1 Department of Clinical Psychology, Psychotherapy, and Experimental Psychopathology, Johannes Gutenberg-University Mainz, Mainz, Germany, 2 Department of Psychology and Psychotherapy, Division Neurobiology and Genetics of Behavior, Witten/Herdecke University, Witten, Germany, 3 Department of Psychology, University of Regina, Regina, Saskatchewan, Canada

* jungmann@uni-mainz.de

**Data Availability Statement:** Dataset used in the research are publicly available on the Open Science Framework at https://osf.io/sm4f2/.

## Abstract

The COVID Stress Scales (CSS) are a new self-report instrument for multidimensional assessment of psychological stress in the context of the pandemic. The CSS have now been translated and validated in over 20 languages, but a validated German version has not yet been available. Therefore, the aim was to develop a German version of the CSS, to test its factor structure, reliability, and validity, and to compare it with international studies. In an online survey (08/2020–06/2021), $N = 1774$ individuals from the German general population (71.5% female; $M_{age} = 41.2$ years, SD = 14.2) completed the CSS as well as questionnaires on related constructs and psychopathology. After eight weeks, participants were asked to participate again for the purpose of calculating retest reliability ($N = 806$). For the German version, the 6-factor structure with good model fit (Root Mean Square Error of Approximation, RMSEA = 0.06) was confirmed, with the six subscales: Danger, Socio-Economic Consequences, Xenophobia, Contamination, Traumatic Stress, and Compulsive Checking. Internal consistencies ranged from ω = .82–.94 (except Compulsive Checking ω = .70), and retest reliability from $r_{tt}$ = .62–.82. Convergent and discriminant validity were confirmed for the German version. Related constructs such as health anxiety, general xenophobia, obsessive-compulsive behavior, and posttraumatic stress disorder symptoms correlated moderately with the respective subscale and lower with the other scales. With anxiety and depression, Traumatic Stress showed the strongest correlation. Overall, there was a high degree of agreement in an international comparison. The CSS can help to identify pandemic-related psychological stress and to derive appropriate interventions.

## Introduction

The COVID-19 pandemic negatively affects the mental health of people worldwide. Pandemic-related stress is multidimensional, involving different domains of life as well as people's emotions, cognitions, and behaviors. Previous studies found that the COVID-19 pandemic is

**Funding:** The authors received no specific funding for this work.

**Competing interests:** The authors have declared that no competing interests exist.

associated with worry and anxiety, hopelessness, pessimism, sleep disturbances, compulsive behaviors, and increased Internet use [1–10]. Meta-analyses showed that there were significant increases in mental health problems in 2020, and some also found significant increases in anxiety and depressive disorders [11]. Studies in Germany also found increased levels of psychological stress, anxiety, depressive symptoms, and symptoms of post-traumatic stress disorder during the pandemic [12–14].

Based on a multidimensional view, Taylor et al. [15, 16] proposed a multifactorial COVID Stress Syndrome that can be assessed using the COVID Stress Scales (CSS). The CSS, a 36 item self-report measure of pandemic-related stress symptoms over the past 7 days, was developed based on symptoms from previous pandemics and current observations and included six subfacets in the original version [16]: Danger (e.g., 'I am worried about catching the virus.'), Socio-Economic Consequences (e.g., 'I am worried about grocery stores running out of cleaning or disinfectant supplies.'), Xenophobia (e.g., 'If I met a person from a foreign country, I'd be worried that they might have the virus.'), Contamination (e.g., 'I am worried that if someone coughed or sneezed near me, I would catch the virus.'), Traumatic Stress (e.g., 'I had trouble sleeping because I worried about the virus.'), and Compulsive Checking/Reassurance (e.g., 'Searched the Internet for treatments for COVID-19.'). A parallel analysis identified five factors, and this 5-factor solution (Danger and Contamination together as one factor) showed good model fit in a second sample. The reliabilities of the CSS subfacets were in the good to very good range, with Cronbach's alpha of .83 to .95, and convergent and discriminant validity were demonstrated [16].

A network analysis showed that worry about danger (Danger subscale) constituted the core characteristic, which was significantly associated with the subfacets Socio-Economic Consequences, Xenophobia, and Traumatic Stress [16]. In addition, Traumatic Stress and Compulsive/Reassurance behaviors were strongly associated. In terms of sociodemographic data, women, individuals with lower levels of education, and unemployed individuals showed higher CSS scores, and significant negative correlations of CSS with age and income were found. The higher health anxiety, anxiety sensitivity, and uncertainty intolerance were before the pandemic, the higher the CSS total score was during the COVID-19 pandemic [15]. In addition, positive associations were found between Covid Stress Syndrome (assessed by CSS) and anxiety symptoms, depressive symptoms, hygiene measures, and beliefs in conspiracy theories [15, 16].

There also seems to be a different vulnerability to the COVID Stress Syndrome depending on the pre-existing mental disorder. Individuals with a pre-existing anxiety disorder showed significantly higher levels on the total score and subscales than individuals with a pre-existing mood disorder and with no mental disorder [except no difference in checking/reassurance between anxiety disorder vs. no mental disorder, 17]. Asmundson et al. [18] examined the COVID Stress Syndrome over the course of the pandemic (early-mid 2020 and early-mid 2021) and associations with current anxiety and mood disorders. CSS scores were higher in the 2020 sample than in the 2021 sample, suggesting a dynamic course of pandemic-related stress. Within mental disorders, individuals with panic disorder showed the highest severity in CSS. Individuals with generalized anxiety disorder, social anxiety disorder, and post-traumatic stress disorder reported, in particular, higher levels of Traumatic Stress.

The CSS has now been translated and validated in over 20 languages, including Dutch, Arabic, Persian, Polish, Serbian, Spanish, and Swedish [19–25]. These versions were found to have a 5- or 6-factor structure, whereby in direct comparisons the 6-factorial model often showed a slightly better model fit [19, 23–25]. The translated versions were able to confirm the reliability and validity of the CSS. Positive correlations of the translated CSS with anxiety and depressive symptoms were equally evident, with the strongest correlations particularly between anxiety

symptoms and the Traumatic Stress subscale [19, 24]. Using the Persian version, Khosravani et al. [22] found that individuals with generalized anxiety disorder, panic disorder, and obsessive-compulsive disorder showed higher levels of COVID Stress Syndrome than individuals with social anxiety disorder and specific phobia.

In Germany, there has also been a lot of research on pandemic-related stress since the beginning of the COVID-19 pandemic. Previous studies showed that the German population suffers from high levels of psychological stress during the pandemic [12, 26]. Stress was measured, for example, with a one-dimensional stress thermometer [27] or the stress module of the Patient Health Questionnaire [28]. However, no validated German version of the CSS is yet available, which can help to assess stress both specifically against the background of the pandemic and multidimensionally. Furthermore, it enables the investigation of the COVID Stress Syndrome in the German population. Therefore, the aim of this study was to translate and validate a German version of the CSS. We expected that the German version to show internal consistencies ($\omega > .80$), retest reliabilities ($r_{tt} > 70$), and validity (correlations with e.g., obsessive-compulsive symptoms, xenophobia) comparable to the English original and the other translations. Since the 5-factor and 6-factor solutions have performed well internationally, these two models were tested and compared for the German version of the CSS.

## Methods

### Data collection procedures and sample

Individuals from the German general population participated in the online study "Stress and Strain during the COVID-19 Pandemic" between August 2020 and June 2021 (recruited via social media, press releases, and flyers). Inclusion criteria were age at least 16 years and written Informed Consent. After eight weeks, participants were invited (via mail) to participate in a second shortened survey (for the purpose of calculating retest reliability of the CSS). For measurement time points 1 and 2, participants could take part in a lottery for a shopping voucher. The study complies with the recommendations of the World Medical Association published in the current version of the Declaration of Helsinki and was approved by the local ethics committee of the Psychological Institute.

Individuals who did not provide written Informed Consent ($N = 118$) and duplicates ($N = 49$) were excluded, resulting in a final sample of $N = 1774$ for measurement time point 1 (main study). The sample was on average M = 41.23 years old (SD = 14.15; range 16–85 years), 71.5% were female, 28.0% male, and 0.5% diverse. In terms of education, half had a college degree (51.1%). Regarding occupational status, 3.1% reported being a pupil/in vocational training, 15.2% a student, 49.3% employed, 14.8% civil servant/self-employed, 6.3% retired, 4.4% househusband/-wife/on parental leave, 1.7% looking for work, and 5.2% other.

The sample at measurement time point 2 (shortened survey eight weeks after the first survey) comprised $N = 852$ individuals (after exclusion of 59 individuals who did not provide written Informed Consent and 6 duplicates). In the case of 46 persons, the individual codes from the 1st and 2nd measurement time points could not be assigned, so that the pooled data set with repeated participation (test and retest after eight weeks) comprised an $n = 806$ ($M_{age} = 42.10$, SD = 14.44 years, 26.6% male, 73.4% female).

### Measures

#### Development of the German version of the CSS

The translation-back-translation process followed the guidelines for translating foreign language self-report measures [29, 30]. The English items were translated into German by the

first and last author and back-translated into English by a professional bilingual translator whose native language is English. The two English versions had only minor differences in wording (e.g., keep me safe–protect me, mail handlers–postman, professionals–experts), but these were retained because of equivalence in content. The German version also consisted of 36 items, six items per scale (see above), and answered on a 5-point Likert scale from 0 = not at all to 4 = extremely (except for the scales Traumatic Stress and Compulsive behavior/reassurance related to frequency: from 0 = never to 4 = almost always). For the final German version of the CSS, see https://coronaphobia.org/professional-resources/.

## Patient health questionnaire-4 [PHQ-4; 31]

The PHQ-4 is an economic self-report measure of depression and anxiety. It was compiled from the two items of the Patient Health Questionnaire-2 [PHQ-2; 32], which inquire about the two core diagnostic criteria of depression, and the two items of the Generalized Anxiety Disorder Screener [GAD-2; 33], which measure the two core criteria of generalized anxiety disorder. Respondents determine symptom severity for the past two weeks on a 4-point Likert scale ranging from not at all (0) to almost every day (3). Reliability and validity have been demonstrated in nonclinical and in clinical samples [31, 34]. In this study, the internal consistency of the PHQ-4 was McDonald's omega $\omega = .88$.

## Short health anxiety inventory [SHAI, 35, 36]

In the present study, the 14-item main scale 'Health anxiety and the feared probability of becoming ill' of the SHAI was used. There are four statements for each item depending on the severity level (0–3; e.g., 0 = 'I do not worry about my health.', . . ., 3 = 'I spend most of my time worrying about my health.'). The SHAI-14 has shown good reliability and validity in previous studies [36, 37]. In this study, the internal consistency was $\omega = .87$.

## Obsessive compulsive inventory-revised [OCI-R, 38, 39]

The OCI-R is an 18-item questionnaire assessing the core symptoms of obsessive-compulsive disorder on six subscales: washing, checking, ordering, obsessing, hoarding, and neutralizing. Respondents indicate on a 5-point Likert scale ranging from not at all (0) to extremely (4) the individual impairment/stress related to the mentioned symptoms for the past month. The OCI-R has been shown to be reliable and valid in nonclinical and clinical samples [38, 40]. For this study, only the washing subscale (e.g., 'I sometimes have to wash or clean myself simply because I feel contaminated.') seems relevant for testing convergent validity. In this study, the internal consistency of the OCI-R was $\omega = .88$.

## Short screening scale for DSM-IV posttraumatic stress disorder [PTSD-Screening; 41, 42]

The PTSD-Screening uses seven items to assess two symptom cluster of PTSD: a) avoidance and numbing (5 items) and b) hyperarousal (2 items). The answers are given on a 4-point Likert scale (1 = not at all to 4 = four times a week/most of the time). The Short Screening-Scale for PTSD is a highly reliable instrument and showed high correlations ($r = .90$, $p < .01$) with symptom scores assessed by a clinical interview [43]. This screening is very efficient with 7 items, covers to a large extent also the DSM-5 criteria, and it was shown that PTSD screenings according to DSM-IV and DSM-5 are largely equivalent [44]. We found an internal consistency of $\omega = .80$ for the PTSD Screening.

### Xenophobia scale [45]

The Xenophobia Scale is a self-report measure for assessing fear-based reactions to strangers. The questionnaire includes nine items (e.g., 'With increased immigration I fear that our way of life will change for the worse.'), which are answered on a 6-point Likert scale from 1 (strongly disagree) to 6 (strongly agree). Van der Veer et al. [45] found reliabilities ranging from $\alpha$ = .77 to $\alpha$ = .86 in a cross-cultural study with American, Norwegian, and Dutch students. In this study, we found an internal consistency of $\omega$ = .95.

All questionnaires were used at measurement time point 1. Measurement time point 2 was a shortened survey including the German version of the CSS and the PHQ-4 (the PHQ-4 at T2 was not evaluated because it was beyond the aim of calculating retest reliability).

### Statistical analyses

The two internationally established models, the original 6- and the 5-factor model of Taylor et al. [16], were tested and compared using a Confirmatory Factor Analysis (CFA, including variance adjusted weighted least squares estimator WLSMV) in Mplus [46]. Regarding model goodness of fit, the Root Mean Square Error of Approximation (RMSEA) was used as the absolute fit index, and the Comparative Fit Index (CFI) and Tucker-Lewis Index (TLI) were used as incremental fit indices. According to Hu and Bentler [47], RMSEA values close to ($\leq$) .06 and CFI and TLI values close to ($\geq$) .95 indicate a good model fit. For the direct comparison of the two models (5- and 6 factor model), the DIFFTEST option in Mplus was used [46]. To estimate reliability, McDonald's omega ($\omega$) using SPSS macro by Hayes and Coutts [48] and test-retest correlations were calculated. Convergent validity was tested via Pearson correlations between the CSS subscales and comparable constructs/corresponding measurement instruments (i.e., Danger, Contamination, and Compulsive Checking subscales and SHAI-14; Xenophobia and Xenophobia Scale; Contamination and OCI-R; Traumatic Stress and PTSD screening). Discriminant validity was examined via comparatively lower correlations of the CSS subscales and substantively more divergent constructs (e.g., Xenophobia subscale and SHAI-14, PTSD, and OCI-R). To statistically compare two correlation coefficients, we used the interactive calculator on the website of Lee and Preacher [49], including Fisher's z transformation. Because it was not possible to skip items in the online studies, the data set did not include missing values.

## Results

### Factor structure

The CFA with six factors indicated a good model fit: $\chi^2(579)$ = 4689.40, p < .001, RMSEA = .063 [90% CI: 0.062–0.065], CFI = .94, TLI = .94. The 5-factor model showed a slightly lower model fit: $\chi^2(584)$ = 6288.58, p < .001, RMSEA = .074 [90% CI: 0.073–0.076], CFI = .92, TLI = .91. The chi-square difference test ($\chi^2(5)$ = 561.47, p < .001) showed a significant difference with superiority of the 6-factor model, so subsequent analyses refer to this 6-factor model (Danger, Socio-Economic Consequences, Xenophobia, Contamination, Traumatic Stress, and Compulsive Checking). Fig 1 shows the results of the CFA with the 6-factor solution.

### Descriptive statistics, inter-correlations, and reliability

Table 1 shows the means, standard deviations, normality (skewness and kurtosis), and internal consistencies (McDonald's $\omega$) for the total CSS and the six subscales. Skewness was between 0.27 and 2.03, and kurtosis between 0.50 and 4.54. The subscales and the total scale are right skewed and show a positive kurtosis, i.e., a peaked distribution (except subscale Danger with a

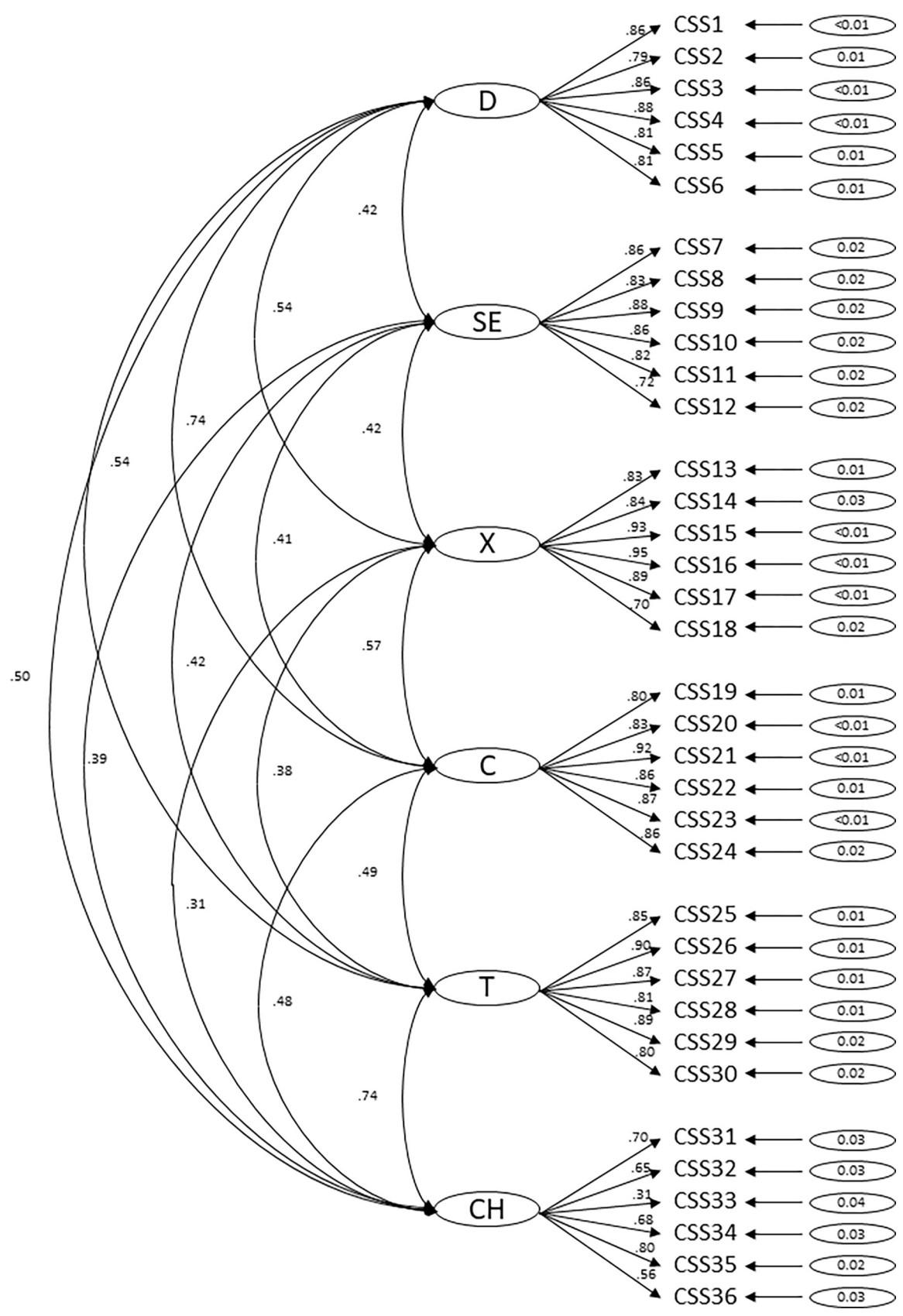

**Fig 1. Six-factor Confirmatory Factor Analysis (CFA) of the German COVID Stress Scales (CSS).** Factor loadings, inter-correlations, and error terms in the Confirmatory Factor Analysis (CFA) of the German version of the COVID Stress Scales (CSS). Subscales D = Danger, SE = Socio-Economic Consequences, X = Xenophobia, C = Contamination, T = Traumatic Stress, CH = Compulsive Checking. CSS1-36 = Items 1 to 36 of the CSS.

negative kurtosis). Bonferroni corrected $t$-tests showed no difference between women and men in the subscales or total scale (corrected $\alpha = .007$, $p \geq .009$, $d \leq 0.14$). With age, small positive correlations were found with CSS total ($r = .07$, $p = .004$), the subscales Socio-Economic Consequences ($r = .13$, $p < .001$), Traumatic Stress ($r = .21$, $p < .001$), and a small negative correlation with Compulsive Checking ($r = -.12$, $p < .001$).

Internal consistencies (McDonald's omega) were $\omega = .94$ (Total), $\omega = .90$ (Danger), $\omega = .82$ (Socio-Economic Consequences), $\omega = .91$ (Xenophobia), $\omega = .88$ (Contamination), $\omega = .89$ (Traumatic Stress), and $\omega = .70$ (Compulsive Checking). Retest reliabilities (test-retest correlations after eight weeks) were $r_{tt} = .82$ (Total), $r_{tt} = .77$ (Danger), $r_{tt} = .70$ (Socio-Economic Consequences), $r_{tt} = .73$ (Xenophobia), $r_{tt} = .77$ (Contamination), $r_{tt} = .71$ (Traumatic Stress), and $r_{tt} = .62$ (Compulsive Checking).

## Convergent and discriminant validity

To confirm convergent validity, moderately strong correlations of the CSS subscales should be found with comparable constructs (Danger, Contamination, and Compulsive Checking and SHAI-14; Xenophobia and Xenophobia Scale; Contamination and OCI-R$_{wash}$; Traumatic Stress and PTSD-Screening). Table 2, confirming convergent validity, shows that these correlations ranged from $r = .37$ (SHAI-14 and Contamination) to $r = .56$ (PTSD and Traumatic Stress). Here, the constructs each showed the strongest correlation with the respective subscale (except for the SHAI-14, with the highest correlation $r = .45$ with Traumatic Stress). There were significantly lower correlations between the constructs (SHAI-14, Xenophobia Scale, OCI-R$_{wash}$, PTSD Screening) and content delineated subscales of the CSS ($Zs \geq 2.59$, $ps \leq .01$), which can be seen as an indication of discriminant validity. Anxiety and depression were included to test discriminant validity. Lower correlations between anxiety and depression and the subscales Danger, Socio-Economic Consequences, Xenophobia, Contamination, and Compulsive Checking were found ($r = .11–.33$), which can be seen as an indication of discriminant validity. Noticeably, both anxiety ($r = .47$) and depression ($r = .40$) had moderate and significantly stronger correlations with Traumatic Stress than with the other subscales ($Zs \geq 5.36$, $ps \leq .001$).

**Table 1. Mean scores, standard deviations, normality (skewness and kurtosis), internal consistencies (McDonald's ω), and (inter)-correlations of the total COVID Stress Scales score (CSS$_{total}$) and the subscales Danger (D), Socio-Economic Consequences (SE), Xenophobia (X), Contamination (C), Traumatic Stress (T), and Compulsive Checking (CH).**

| Scale | M | SD | Skewness | Kurtosis | ω | 1 | 2 | 3 | 4 | 5 | 6 |
|---|---|---|---|---|---|---|---|---|---|---|---|
| CSS$_{total}$ | 27.55 | 17.91 | 1.11 | 1.63 | .94 | .82 | .58 | .71 | .80 | .70 | .61 |
| 1. D | 8.54 | 5.40 | 0.27 | -0.58 | .90 | | .35 | .48 | .65 | .45 | .36 |
| 2. SE | 2.32 | 3.21 | 2.03 | 4.54 | .82 | | | .35 | .34 | .33 | .27 |
| 3. X | 4.75 | 4.70 | 1.16 | 1.17 | .91 | | | | .52 | .32 | .24 |
| 4. C | 6.05 | 4.58 | 0.88 | 0.50 | .88 | | | | | .41 | .36 |
| 5. T | 2.94 | 3.79 | 1.84 | 3.62 | .89 | | | | | | .56 |
| 6. CH | 2.96 | 3.23 | 1.49 | 2.58 | .70 | | | | | | |

all $ps < .001$.

**Table 2. Pearson correlations of the total score of CSS and the subscales and health anxiety (SHAI-14), general xenophobia (XP Scale), PTSD symptoms (Short screening scale for DSM-IV posttraumatic stress disorder), compulsive washing (OCI-R$_{wash}$), anxiety (PHQ-4), and depression (PHQ-4).**

|  | CSS$_{total}$ | D | SE | X | C | T | CH |
|---|---|---|---|---|---|---|---|
| SHAI-14 | .48 | .38 | .23 | .29 | .37 | .45 | .36 |
| XP Scale | .28 | .08 | .29 | .49 | .09 | .15 | .09 |
| OCI-R$_{wash}$ | .41 | .26 | .24 | .25 | .40 | .32 | .28 |
| PTSD | .49 | .36 | .23 | .25 | .32 | .56 | .41 |
| Anxiety | .36 | .25 | .18 | .15 | .23 | .47 | .33 |
| Depression | .30 | .22 | .13 | .11 | .19 | .40 | .29 |

D = Danger, SE = Socio-Economic Consequences, X = Xenophobia, C = Contamination, T = Traumatic Stress, CH = Compulsive Checking subscale of the CSS. All ps < .001.

## Discussion

Our study aimed to translate and validate a German version of the CSS. Although the CSS has been validated in more than 20 languages, the German version has not yet been tested.

For the German version of the CSS, the 6-factor model showed a significantly better model fit than the 5-factor model. This is consistent with the originally developed first English version [16] and other translated versions [Arabic, Polish, Spanish, and Swedish versions, 19, 21, 24, 25] which illustrates the international comparability. Taylor et al. [16] found five factors in their parallel analysis and summarized the scales Danger and Contamination. In the German version, these subscales showed a high and the highest intercorrelation (*r* = .65), so that the German version also confirms their proximity to each other. Nevertheless, the two subscales seem to be sufficiently differentiated from each other [24] and international studies found in direct comparisons a superiority of the 6-factor solution over the 5-factor solution [19, 23–25]. Possible explanations for this slight deviation in factor structure could be due to the dynamic events of the COVID-19 pandemic as well as country-specific incidences and measures (e.g., change in threat due to changes in incidence, increase in knowledge regarding the virus, changes in protective measures). For example, the survey in the general population by Taylor et al. [16] occurred earlier (03/04 2020) than ours and studies in other countries (08/2020-06/2021), with possibly country-specific education and protective measures in addition to divergent incidences.

The means of the scale sums are comparable to other translated versions used in the general population (when means are reported), they are descriptively between, for example, the Swedish version (descriptively slightly lower values) [24] and the Serbian version (mostly descriptively slightly higher values) [23]. In the English version [data collection 03-04/2020, 16], values were slightly higher descriptively, whereas it was also shown in a longitudinal design [18] that CSS values were higher in an earlier phase of the pandemic (03-04/2020) than in a later phase (03-05/2021). We found, similar to Carlander et al. [24] and Mahamid et al. [21], no sex differences in the CSS, whereas Taylor et al. [15] found a significantly higher total score for women. It is possible that this could also be due to the early measurement period in Taylor et al. [15] as previous studies found that women in particular were stressed at the beginning of the pandemic and stress decreased over time [50]. Similar to Carlander et al. [24] we found a very small positive correlation between age and CSS total, whereas at the subscale level we found in particular a significant positive correlation with Xenophobia. This is consistent with findings that older people tend to have more negative attitudes toward immigrants [although in cross-sectional studies age and cohort effects may be present, 51]. The small, but significant, negative correlation, between age and the Compulsive Checking subscale could be explained

by the fact that half of the items involve Internet use (i.e., You Tube and posts on social media).

The inter-correlations with the three highest correlations between Danger and Contamination (r = .65), between Traumatic Stress and Compulsive Checking (r = .56), and between Danger and Xenophobia (r = .48) are consistent with the network analysis of Taylor et al. [15] who found the strongest interconnections between Traumatic Stress and Compulsive Checking, between Danger/Contamination (factor summarized here) and Xenophobia, as well as between Danger/Contamination and Socio-Economic Consequences.

Internal consistencies were in the good to very good range (ω = .82–.94), except for the Compulsive Checking scale, which had an acceptable internal consistency (ω = .70). Also in international studies, Compulsive Checking was found to have the lowest consistency of the subscales, e.g., Carlander et al. [24] also found a value of ω = .72. Possibly, this could be due to the fact that the scale includes both, items describing a more general search for information (e.g., 'YouTube videos about COVID-19.') and items with behaviors related to health concerns or with the intention to reduce them (e.g., 'Seeking reassurance from friends or family about COVID-19.'). The source of the information search could also play a role. If someone does not use You Tube or social media, these two items would be negated per se. Using the German version of the CSS, we found quite high stability of COVID stress syndrome over eight weeks, with the behavioral scale Compulsive Checking showing the lowest stability ($r_{tt}$ = .62) and worries about getting infected and that measures are not enough (Danger) and Contamination being the most stable ($r_{tt}$ = .77). The retest reliability seems to be similar internationally, e.g., Mahamid [21] found a correlation of $r_{tt}$ = .81 (after three weeks) for the total scale of the Arabic version (German version $r_{tt}$ = .82).

The German version was also able to show its validity. The subscales each correlated significantly and moderately strongly with related constructs. As in previous studies [16, 24], relevant positive associations (r >.30) were found between health anxiety (SHAI-14) and Danger, Contamination, Compulsive Checking, and especially Traumatic Stress. Regarding the latter, the stronger the health concerns, the stronger the hyperarousal and intrusive experience related to the COVID-19 pandemic. This is also consistent with the finding of Asmundson et al. [18] that patients with panic disorder who have health concerns related to an acute physical incident are particularly vulnerable to COVID stress syndrome. The role of health anxiety and the associated trait of uncertainty intolerance in COVID stress syndrome (CSS) was also highlighted by the study of Taylor et al. [52]. They found that the association between negative affectivity and CSS is mediated by uncertainty intolerance and tendency toward health anxiety. Regarding xenophobia and compulsive washing, the strongest correlations were with the respective subscale of the CSS (Xenophobia and Contamination) and also comparable in strength to previous studies [16, 19]. The overlap of CSS (especially Traumatic Stress) with symptoms of PTSD has been hypothesized [53], but previous validations have rarely used a corresponding PTSD scale. With the German version we could confirm the very good validity of the subscale Traumatic Stress (with PTSD screening r = .56, with other subscales $r \leq$ .41). Generalized anxiety and depression (PHQ-4) were recorded primarily to test discriminate validity. Apart from medium-strong correlations with the Traumatic Stress subscale (r = .47/ .40), there are lower correlations with the other subscales (r = .11–.33), which is also consistent with further translations of the CSS [19, 24]. The higher correlation between Traumatic Stress and, in particular, generalized anxiety can be explained by the similarity in content of the items dealing with anxiousness/tension (compare hyperarousal) and difficulties in controlling cognitions (compare intrusive experiencing).

Practical implications of the CSS are its use both, in the general population and in individuals with mental disorders, to assess the level of pandemic-related psychological burden and

thus the need for interventions (e.g., to prevent or reduce the development or worsening of mental distress or disorders). In addition to the general need for intervention, the CSS can be used to derive the exact need for specific interventions either at the individual or at the population level. For example, mean scores were highest for the Danger and Contamination subscales. Here, political health campaigns to educate, for example, the transmission routes could be helpful, or in the case of patients with a mental disorder, corresponding interventions similar to those for the treatment of anxiety and obsessive-compulsive disorders [17, 18, 22]. In addition, more innovative approaches should also be encouraged, such as promoting a life span perspective (e.g., differentiated consideration of different age groups and the transition age), more targeted programs (e.g., families, health care workers), concrete formats such as promoting telehealth (i.e., audio/video), brief population-based/public prevention and interventions, and lay-provider systems [54].

Some limitations should be mentioned. Our sample consisted of more than 2/3 women, was comparatively young and educated, thus not representative. These sample characteristics (unequal distribution of gender and younger) could also (partly) explain the slight deviations from the study of Taylor et al. [15] regarding the correlations of CSS with sociodemographic variables. For a validation study and against the background of a dynamic pandemic, the time period of the survey was quite long (08/2020-06/2021). Although some study confirmed factorial invariance over time [18, 19], the dynamic happening and, for example, changes in mean values over time might have affected comparability with the English version and other translations.

## Conclusions

In summary, the German version of the CSS is a reliable and valid method to assess psychological stress during pandemics multifactorially. The German version comprises six subfacets with the scales: Danger, Socio-Economic Consequences, Xenophobia, Contamination, Traumatic Stress, and Compulsive Checking. Internationally, there is a high degree of agreement regarding factor solution, psychometric quality, and correlations with related constructs as well as psychopathology. The CSS was developed for immediate application in the context of the current pandemic but, also, to be adaptable for application in future pandemics; as such, the German CSS adds another translation to the corpus of available scales and positions researchers to assess both current and future pandemic-stress using the current gold standard.

## Acknowledgments

**Acknowledgments** We thank Steven Stelz for assistance with the literature review and description of some measurement instruments.

## Author Contributions

**Conceptualization:** Stefanie M. Jungmann, Michael Witthöft.

**Data curation:** Stefanie M. Jungmann, Vincent Nin.

**Formal analysis:** Stefanie M. Jungmann, Vincent Nin.

**Investigation:** Stefanie M. Jungmann, Martina Piefke.

**Methodology:** Stefanie M. Jungmann.

**Project administration:** Stefanie M. Jungmann, Martina Piefke.

**Validation:** Stefanie M. Jungmann.

**Visualization:** Stefanie M. Jungmann.

**Writing – original draft:** Stefanie M. Jungmann.

**Writing – review & editing:** Stefanie M. Jungmann, Martina Piefke, Vincent Nin, Gordon J. G. Asmundson, Michael Witthöft.

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
