## [Decision Letter · Decision Letter 0]

1 Dec 2022

PONE-D-22-28279COVID-19 Stress Syndrome in the German general population: Validation of a German version of the COVID Stress ScalesPLOS ONE

Dear Dr. Jungmann,

Thank you for submitting your manuscript to PLOS ONE. After careful consideration, we feel that it has merit but does not fully meet PLOS ONE’s publication criteria as it currently stands. Therefore, we invite you to submit a revised version of the manuscript that addresses the points raised during the review process.

One expert reviewer in the field of psychometrics has reviewed the work and provided some comments for revision. I agree with the opinions from the reviewer that the present contribution in general is good. However, it needs to be revised before publication. Apart from the reviewer's comments, the authors should consider the following of my concerns as well.1. I think that the authors did not provide a figure legend to explain their figure 1. Specifically, the authors have used some abbreviations in the figure 1 and their figure legend should explain these abbreviations because figure should stand alone.2. The authors mentioned the abbreviation of RMSEA in the Abstract without providing a full spell-out. Given that Abstract also stands alone to the main text, the full spell-out of the RMSEA should be given in the Abstract.3. This may be a stylish issue; however, I prefer the authors report the full spell-out of the abbreviations and put the abbreviations in the brackets. Instead of reporting the abbreviations and putting the full spell-outs in the brackets.4. The Introduction has used some systematic reviews to describe the mental health issues during COVID-19; however, I would encourage the authors further using the following systematic reviews and large-scale studies to strengthen the descriptions.Khankeh H, Pourebrahimi M, Karibozorg MF, Hosseinabadi-Farahani M, Ranjbar M, Ghods MJ, Saatchi M. Public trust, preparedness, and the influencing factors regarding COVID-19 pandemic situation in Iran: A population-based cross-sectional study. Asian J Soc Health Behav 2022;5:154-61Vicerra PM. Mental stress and well-being among low-income older adults during COVID-19 pandemic. Asian J Soc Health Behav 2022;5:101-7Hasannia E, Mohammadzadeh F, Tavakolizadeh M, Davoudian N, Bay M. Assessment of the anxiety level and trust in information resources among iranian health-care workers during the pandemic of coronavirus disease 2019. Asian J Soc Health Behav 2021;4:163-8Olashore AA, Akanni OO, Fela-Thomas AL, Khutsafalo K. The psychological impact of COVID-19 on health-care workers in African Countries: A systematic review. Asian J Soc Health Behav 2021;4:85-97Rajabimajd N, Alimoradi Z, Griffiths MD. Impact of COVID-19-related fear and anxiety on job attributes: A systematic review. Asian J Soc Health Behav 2021;4:51-55. This statement "prove to be as reliable (ω > .80)" should be revised. Although many people consider that internal consistency is a type of reliability; its nature is more like validity (i.e., all items are associated with each other). Therefore, I would use test-retest reliability to emphasize if the tested instrument (i.e., CSS) is reliable. The hypothesis of ω here is fine, but it would be much better if test-retest reliability is also hypothesized.6. I prefer the authors not using some strong words (e.g., prove) in the present submission.7. As the present sample included some participants aged below 18 years, I wonder if the authors have obtained their parents' informed consent. 8. The authors did a chi-square difference test; however, this is not mentioned in their Statistical analysis section. Although some people know that the DIFFTEST option in Mplus is used for chi-square difference test, not everybody knows this. Please clearly mention this in the Statistical analysis section.9. The authors have compared the correlation coefficients; however, this is not stated in the Statistical analysis section either. I think that the authors have used Lee & Preacher's website to make the correlation coefficient comparisons. However, they did not disclose this. Or, if I get it wrong, the authors should provide a proper citation to indicate how they compared the correlation coefficients. 

We look forward to receiving your revised manuscript.

Kind regards,

Chung-Ying Lin

Academic Editor

PLOS ONE

Journal Requirements:

Reviewers' comments:

Reviewer's Responses to Questions

**Comments to the Author**

1. Is the manuscript technically sound, and do the data support the conclusions?

Reviewer #1: Yes

2. Has the statistical analysis been performed appropriately and rigorously? 

Reviewer #1: Yes

3. Have the authors made all data underlying the findings in their manuscript fully available?

Reviewer #1: Yes

4. Is the manuscript presented in an intelligible fashion and written in standard English?

Reviewer #1: Yes

5. Review Comments to the Author

Reviewer #1: Thank you for the opportunity to review this work. The authors attempted to validate the German version of COVID Stress Scales. The validation would be advantageous to the target population. However, there are a few small issues that need to be addressed.

1. Line 115, it is more appropriate to say “translate and validate” rather than “develop and validate”.

2. Lines 142-143, what is the meaning of "denied the inclusion criteria"?

3. Line 146, “26.6% m, 73.4% f”. Please use “male” and “female”.

4. Please provide the reliability of PHQ-4, SHAI, OCI-R(wash), PTSD-Screening, Xenophobia Scale in current study.

5. Line 203, What is the purpose of PHQ-4 in time point 2?

6. In statistical analyses, description about convergent and discriminant validity was missing.

7. In results, it would be better to report “descriptive statistics, inter-correlations, and reliability” before “Factor structure”.

8. Line 289, again, it is more appropriate to say “translate and validate” rather than “develop and validate”.

9. Lines 310–312, the use of square brackets may confuse readers. For example, [descriptively slightly lower values, 20], readers may think 20 is the value, but it is the citation. Use round bracket is fine, i.e. (descriptively slightly lower values) [20].

6. PLOS authors have the option to publish the peer review history of their article (what does this mean?). If published, this will include your full peer review and any attached files.

Reviewer #1: No

---

## [Author Response · Author response to Decision Letter 0]

2 Dec 2022

Dear Professor Lin,

Dear reviewer,

Thank you very much for your letter, and for the very helpful review of our manuscript titled "COVID-19 Stress Syndrome in the German general population: Validation of a German version of the COVID Stress Scales".

We thank you and the reviewer for the thoughtful and constructive feedback and suggestions for improving the manuscript. We have revised the manuscript according to your recommendations. We have carefully considered and responded in detail to each of the points made by you and the reviewer. Please find our actions detailed in the following:

Comments editor:

One expert reviewer in the field of psychometrics has reviewed the work and provided some comments for revision. I agree with the opinions from the reviewer that the present contribution in general is good. However, it needs to be revised before publication. Apart from the reviewer's comments, the authors should consider the following of my concerns as well.

1. I think that the authors did not provide a figure legend to explain their figure 1. Specifically, the authors have used some abbreviations in the figure 1 and their figure legend should explain these abbreviations because figure should stand alone.

[AU]: Thank you for the thoughtful comment. We have added a legend for Figure 1 on page 9 in the revised manuscript.

“Factor loadings, inter-correlations, and error terms in the Confirmatory Factor Analysis (CFA) of the German version of the COVID Stress Scales (CSS). Subscales D = Danger, SE = Socio-Economic Consequences, X = Xenophobia, C = Contamination, T = Traumatic Stress, CH = Compulsive Checking. CSS1-36 = Items 1 to 36 of the CSS.” (page 9)

2. The authors mentioned the abbreviation of RMSEA in the Abstract without providing a full spell-out. Given that Abstract also stands alone to the main text, the full spell-out of the RMSEA should be given in the Abstract.

[AU]: We have RMSEA spelled out in the abstract. Since the abbreviation is very common, we have left it additional.

“(Root Mean Square Error of Approximation, RMSEA = 0.06)”

3. This may be a stylish issue; however, I prefer the authors report the full spell-out of the abbreviations and put the abbreviations in the brackets. Instead of reporting the abbreviations and putting the full spell-outs in the brackets.

[AU]: Thank you for pointing this out. We have adopted the changes as recommended, first written out, then the abbreviation in brackets.

“Regarding model goodness of fit, the Root Mean Square Error of Approximation (RMSEA) was used as the absolute fit index, and the Comparative Fit Index (CFI) and Tucker-Lewis Index (TLI) were used as incremental fit indices.” (page 8)

4. The Introduction has used some systematic reviews to describe the mental health issues during COVID-19; however, I would encourage the authors further using the following systematic reviews and large-scale studies to strengthen the descriptions.

Khankeh H, Pourebrahimi M, Karibozorg MF, Hosseinabadi-Farahani M, Ranjbar M, Ghods MJ, Saatchi M. Public trust, preparedness, and the influencing factors regarding COVID-19 pandemic situation in Iran: A population-based cross-sectional study. Asian J Soc Health Behav 2022;5:154-61

Vicerra PM. Mental stress and well-being among low-income older adults during COVID-19 pandemic. Asian J Soc Health Behav 2022;5:101-7

Hasannia E, Mohammadzadeh F, Tavakolizadeh M, Davoudian N, Bay M. Assessment of the anxiety level and trust in information resources among iranian health-care workers during the pandemic of coronavirus disease 2019. Asian J Soc Health Behav 2021;4:163-8

Olashore AA, Akanni OO, Fela-Thomas AL, Khutsafalo K. The psychological impact of COVID-19 on health-care workers in African Countries: A systematic review. Asian J Soc Health Behav 2021;4:85-97

Rajabimajd N, Alimoradi Z, Griffiths MD. Impact of COVID-19-related fear and anxiety on job attributes: A systematic review. Asian J Soc Health Behav 2021;4:51-5

[AU]: Thank you for the recommendations regarding other interesting studies for our theoretical background. We have now cited 4 of these studies as suggested, which we think fit well with our theoretical background (Hasannia et al., 2021; Manalang Vicerra, 2022; Olashore et al., 2021; Rajabimajd et al., 2021). Regarding the sentence with the meta-analysis, we had specifically aimed at a before vs. during comparison (increase during the pandemic).

5. This statement "prove to be as reliable (ω > .80)" should be revised. Although many people consider that internal consistency is a type of reliability; its nature is more like validity (i.e., all items are associated with each other). Therefore, I would use test-retest reliability to emphasize if the tested instrument (i.e., CSS) is reliable. The hypothesis of ω here is fine, but it would be much better if test-retest reliability is also hypothesized.

[AU]: Thank you for this helpful hint, we have reworded this sentence and added a retest reliability hypothesis (on page 5)

“We expected that the German version to show internal consistencies (ω > .80), retest reliabilities (rtt > 70), and validity (correlations with e.g., obsessive-compulsive symptoms, xenophobia) comparable to the English original and the other translations.” (on page 5)

6. I prefer the authors not using some strong words (e.g., prove) in the present submission.

[AU]: As recommended, we have replaced strong words, especially "proven" in the complete manuscript.

7. As the present sample included some participants aged below 18 years, I wonder if the authors have obtained their parents' informed consent. 

[AU]: Thank you for your inquiry regarding the participation of 16 and 17 year old participants. In the present study, participation was possible from the age of 16. Separate parental consent for 16 and 17 year olds was not required. The questions we asked are appropriate from the age of 16. Within the framework of the European Union's General Data Protection Regulation, processing and storage of personal data (anonymous survey in this case) is possible from the age of 16. This procedure has been reviewed and positively assessed by the ethics committee.

8. The authors did a chi-square difference test; however, this is not mentioned in their Statistical analysis section. Although some people know that the DIFFTEST option in Mplus is used for chi-square difference test, not everybody knows this. Please clearly mention this in the Statistical analysis section.

[AU]: Thank you for pointing this out. We had briefly mentioned the DIFF test in the statistical analyses, but have emphasized this more clearly in the now revised version.

“For the direct comparison of the two models (5- and 6 factor model), the DIFFTEST option in Mplus was used [46].” (page 9)

9. The authors have compared the correlation coefficients; however, this is not stated in the Statistical analysis section either. I think that the authors have used Lee & Preacher's website to make the correlation coefficient comparisons. However, they did not disclose this. Or, if I get it wrong, the authors should provide a proper citation to indicate how they compared the correlation coefficients. 

[AU]: Thank you also for this comment. We agree that this is a relevant point. We have now added a supplementary description including the reference in the statistical analysis section (page 9)

“To statistically compare two correlation coefficients, we used the interactive calculator on the website of Lee and Preacher [49], including Fisher's z transformation.” (Page 9)

Comments external reviewer:

Reviewer #1: Thank you for the opportunity to review this work. The authors attempted to validate the German version of COVID Stress Scales. The validation would be advantageous to the target population. However, there are a few small issues that need to be addressed.

1. Line 115, it is more appropriate to say “translate and validate” rather than “develop and validate”.

[AU]: Thank you for pointing this out. As recommended, the revised version includes "translate and validate" (page 5).

2. Lines 142-143, what is the meaning of "denied the inclusion criteria"?

[AU]: These subjects had accessed the questionnaire but had not agreed to the consent form ("I hereby consent to participate voluntarily in the study....."). For better understanding, we have chosen a different wording here. (page 6)

“The sample at measurement time point 2 (shortened survey eight weeks after the first survey) comprised N = 852 individuals (after exclusion of 59 individuals who did not provide written Informed Consent and 6 duplicates).” (page 6)

3. Line 146, “26.6% m, 73.4% f”. Please use “male” and “female”.

[AU]: In the revised version of the manuscript, we have now used "male" and "female".

4. Please provide the reliability of PHQ-4, SHAI, OCI-R(wash), PTSD-Screening, Xenophobia Scale in current study.

[AU]: Thank you for this important advice. We have added the internal consistency in each case.

PHQ-4: “In this study, the internal consistency of the PHQ-4 was McDonald's omega ω = .88.” (Page 7)

SHAI-14: “In this study, the internal consistency was ω = .87.” (page 7)

OCI-R: “In this study, the internal consistency of the OCI-R was ω = .88.” (page 8)

PTSD-Screening: “We found an internal consistency of ω = .80 for the PTSD Screening.” (page 8)

Xenophobie Scale: “In this study, we found an internal consistency of ω = .95.” (page 8)

5. Line 203, What is the purpose of PHQ-4 in time point 2?

[AU]: Thank you for the comment, that is a very understandable query. Originally, we had the intention to look at correlations between the COVID Stress Scales and anxiety and depression at T2 as well. However, it was beyond the aim and scope of this validation study. In the revised version, we have explained this for better comprehensibility.

“Measurement time point 2 was a shortened survey including the German version of the CSS and the PHQ-4 (the PHQ-4 at T2 was not evaluated because it was beyond the aim of calculating retest reliability).” (page 8)

6. In statistical analyses, description about convergent and discriminant validity was missing.

[AU]: Thank you very much for this suggestion. We have now added the description of convergent and discriminant validity in the statistical methods section.

“Convergent validity was tested via Pearson correlations between the CSS subscales and comparable constructs/corresponding measurement instruments (i.e., Danger, Contamination, and Compulsive Checking subscales and SHAI-14; Xenophobia and Xenophobia Scale; Contamination and OCI-R; Traumatic Stress and PTSD screening). Discriminant validity was examined via comparatively lower correlations of the CSS subscales and substantively more divergent constructs (e.g., Xenophobia subscale and SHAI-14, PTSD, and OCI-R).” (page 9)

7. In results, it would be better to report “descriptive statistics, inter-correlations, and reliability” before “Factor structure”.

[AU]: When writing the results section, we had exactly the same thoughts. We then decided on this order for the following reason: The result of the factor structure (5 or 6 factors) influences the scale formation, which is the basis of the descriptive statistics, intercorrelation and reliability per subscale, i.e. whether 5 or 6 subscales are considered in more detail. We hope our explanation of the order is comprehensible and finds support.

8. Line 289, again, it is more appropriate to say “translate and validate” rather than “develop and validate”.

[AU]: We have also replaced "develop" with "translate" at this point (in the revised version of article line 306).

9. Lines 310–312, the use of square brackets may confuse readers. For example, [descriptively slightly lower values, 20], readers may think 20 is the value, but it is the citation. Use round bracket is fine, i.e. (descriptively slightly lower values) [20].

[AU]: Thank you for this recommendation to make the information in the brackets clearer. We have changed this as follows:

“The means of the scale sums are comparable to other translated versions used in the general population (when means are reported), they are descriptively between, for example, the Swedish version (descriptively slightly lower values) [24] and the Serbian version (mostly descriptively slightly higher values) [23].” (page 14)

Yours sincerely,

Stefanie Jungmann

---

## [Editor Report · Decision Letter 1]

5 Dec 2022

COVID-19 Stress Syndrome in the German general population: Validation of a German version of the COVID Stress Scales

PONE-D-22-28279R1

Dear Dr. Jungmann,

We’re pleased to inform you that your manuscript has been judged scientifically suitable for publication and will be formally accepted for publication once it meets all outstanding technical requirements.

Kind regards,

Chung-Ying Lin

Academic Editor

PLOS ONE

Additional Editor Comments (optional):

I have evaluated the authors' responses to the reviewer's comments and my comments. The authors have satisfactorily addressed all the comments and improve the manuscript satisfactorily. I applaud and thank for the authors' efforts in this contribution and am happy to accept this contribution. 
---

## [Editor Report · Acceptance letter]

24 Jan 2023

PONE-D-22-28279R1 

COVID-19 Stress Syndrome in the German general population: Validation of a German version of the COVID Stress Scales 

Dear Dr. Jungmann:

I'm pleased to inform you that your manuscript has been deemed suitable for publication in PLOS ONE. Congratulations! Your manuscript is now with our production department. 

Kind regards, 

on behalf of

Dr. Chung-Ying Lin 

Academic Editor

PLOS ONE